# Extraction of Polysaccharides from Root of *Pseudostellaria heterophylla* (Miq.) Pax. and the Effects of Ultrasound Treatment on Its Properties and Antioxidant and Immune Activities

**DOI:** 10.3390/molecules29010142

**Published:** 2023-12-26

**Authors:** Hangyu Li, Ziwei Liu, Qianqian Liu, Xinnan Zhang, Sheng Li, Feng Tang, Linzi Zhang, Qian Yang, Qiran Wang, Shuyao Yang, Ling Huang, Yuwei Ba, Xihui Du, Falong Yang, Haibo Feng

**Affiliations:** 1College of Animal Husbandry and Veterinary Medicine, Southwest Minzu University, Chengdu 610041, China; li1998hangyu@163.com (H.L.); z774513726@126.com (Z.L.); 15892603728@163.com (Q.L.); n1547187363@163.com (X.Z.); listen699@163.com (S.L.); tangfeng1719556170@163.com (F.T.); zlz754130837@163.com (L.Z.); qianq1103@163.com (Q.Y.); w2329839437@163.com (Q.W.); carry_young@163.com (S.Y.); hl17723661622@163.com (L.H.); bayuwei1998@163.com (Y.B.); duxihui2022@163.com (X.D.); 2Key Laboratory of Ministry of Education and Sichuan Province for Qinghai—Tibetan Plateau Animal Genetic Resource Reservation and Utilization, Chengdu 610041, China

**Keywords:** *Pesudostellariae radix*, polysaccharides, ultrasound, antioxidant, immune

## Abstract

The hydrophilic polysaccharides (PS) were isolated and purified from the tuberous roots of *Pseudostellaria heterophylla*. The extraction process of PS from *Pesudostellariae radix* was optimized by single–factor experiments and orthogonal design. The extract was purified by DEAE cellulose column to obtain the pure polysaccharide PHP. Then PHP was treated with different intensities of sonication to study the effect of sonication on PHP’s characteristics and its biological activity in vitro and in vivo. The results of this study revealed that ultrasound treatment did not significantly change the properties of PHP. Further, with the increase of ultrasound intensity, PHP enhanced the proliferation and phagocytosis of macrophage RAW264.7. Meanwhile, it could also significantly improve the body’s antioxidant activity and immune function. The results of this study demonstrated that PHP has the potential as a food additive with enhanced antioxidant and immune functions, and its biological activities could be enhanced by sonication.

## 1. Introduction

*Pesudostellariae radix* (Tai Zi Shen), also known as Hai Er Shen [1], is the dried root of *Pseudostellaria heterophylla* (Caryophyllaceae) and is widely used in traditional Chinese medicine (TCM) due to its beneficial health effects [2]. Wild *Pseudostellariae heterophylla* is an herbaceous perennial with characteristics of stem–rooting and swelled roots. It has great potential for use in healthcare products. Rapid production of this plant has been achieved through domestication and cultivation, with a production cycle of only 7–8 months. Currently, Guizhou Shibing, Anhui Xuanzhou, and Fujian Rongrong are the main production areas. *P. radix*’s usage was first described in Wu Yiluo’s ‘Ben Cao Cong Xin’ in the Qing Dynasty. It reinforces the spleen, replenishes Qi, and moistens the lungs [3,4]. It has been reported that *P. radix* has a wide range of biological activities and contains various chemical components, including polysaccharides, glycosides, cyclic peptides, sterols, oils, and other volatile oily substances [5].

Plant polysaccharide (PS) is a macromolecular polymer with a relatively complex structure, consisting of more than a dozen monosaccharides such as mannose, galactose, glucose, and arabinose connected by glycosidic bonds in the form of straight or branched chains [6,7,8]. PSs are one of the most active substances in plants and play a key role in immune regulation [9]. PS has also been reported to be used in the treatment of elevated blood lipids and also as having anticancer [10], antioxidant [11], antiviral [12,13], antidiabetic [14], and immuno–adjuvant properties [15,16].

In everyday life activities, oxidation is vital for providing energy. However, when there is a problem in the metabolic pathway, the body’s antioxidant system cannot eliminate the excess oxygen free radicals, damaging tissues and cells [17,18]. Recent studies have shown that bio–polysaccharides isolated from *Ganoderma capense*, *Phyllostachys pubescens* [19], and *Chuanminshen violaceus* [20] have strong antioxidant qualities and efficiently scavenge free radicals [21].

In PS extraction, the use of ultrasound is expected to impact PS’s physicochemical properties and characterization, since ultrasound has been reported to improve the solubility and diffusivity of PS [22]. This study aimed to optimize the extraction of PS from *P. radix* and determine the macrophage RAW264.7 activity, phagocytosis, NO production, and in vivo antioxidant activity of ultrasonicated *P. radix* PS. The results of this study will add to the existing knowledge about *P. radix* PS and build a foundation for further applications of PS as a natural antioxidant and immunopotentiator.

## 2. Results and Discussion

### 2.1. Purification of PS

From the orthogonal test results (Appendix A, Appendix A), the scheme with the highest extraction rate was selected to extract crude polysaccharide PS. The extracted PS was purified by DEAE cellulose–s28556 column. As shown in Figure 1, the PS content was the largest in the first 40 tubes eluted with distilled water. The eluted polysaccharide content gradually decreased with the increase in NaCl concentration. Therefore, the first 40 tubes of eluate were collected, concentrated, and lyophilized to obtain the purified polysaccharide PHP.

### 2.2. Characterization of PHP

#### 2.2.1. FTIR Analysis

The FTIR spectrum of PHP was recorded in the 4000–400 cm^−1^ range. According to Figure 2A, the FTIR spectrum PHP matched the characteristics of the absorption spectra of polysaccharides. There was an intensive, broad absorption peak at 3275 cm^−1^ corresponding to PHP’s intermolecular, hydrogen–bonded OH groups. The spectrum also showed a low–intensity peak at 2923 cm^−1^ attributed to C–H stretching vibrations [23]. The peak at 1631 cm^−1^ was the bending vibration of O–H, while 1411 cm^−1^ and 1353 cm^−1^ were the absorption peaks generated by the =CH2 deformation and C–H bending, respectively [24]. Interestingly, the absorption peak at 1145 cm^−1^ was assigned to the symmetrical tensile vibration of the S=O bond. The peak at 848 cm^−1^ demonstrated the presence of an α–glycoside bond [25]. Thus, the FTIR analysis indicated that PHP could be classified as a sulfate–containing polysaccharide.

#### 2.2.2. Gel Permeation Chromatography Spectrum

The GPC spectrum of PHP showed two main peaks (Figure 2B). The molecular weight (M_w_) of the PHP peaks was determined by GPC. The number average M_w_ of Peak1 PHP was 463,500 g/mol, while Peak 2 PHP weighed 125,900 g/mol. The corresponding weighted average, M_w_, are 1,159,000 and 168,900 g/mol, respectively. The distribution index (PD) is an indicator of M_w_ distribution. The wider the M_w_ distribution, the larger the PD value. The PD values closer to 1 indicated narrower M_w_ distribution. As can be seen from Table 1, the M_w_ distribution of PHP corresponding to Peak 2 was more uniform than Peak 1.

#### 2.2.3. Monosaccharide Composition of *P. radix* PHP

The monosaccharide content of PHP was determined by LC after acid hydrolysis and pre–column derivatization (Figure 2C). The monosaccharides were identified by the retention times of corresponding monosaccharide standards. Galactose, glucose, arabinose, mannose, L–rhamnose monohydrate, galacturonic acid, xylose, glucuronic acid, fucose, and ribose with the molar ratios of 743.54, 718.78, 118.1, 20.91, 12.05, 9.06, 4.78, 4.77, 3.93, and 1.19 were the main constituents of PHP. Among them, galactose, glucose, and arabinose had the highest content, respectively, and the result was similar to the monosaccharide composition of *Ixeris polycephala* polysaccharide [26], which showed good immune activity.

### 2.3. Characterization of Ultrasound PHP

#### 2.3.1. IR Analysis

The IR spectra of PHP, PHP1, and PHP2 are shown in Figure 3A. From the figure, we can see that compared with PHP, the IR spectra of PHP1 and PHP2 were left shifted, but there was no obvious change in the functional groups, indicating that ultrasound treatment had no significant effect on the functional groups of PHP [27].

#### 2.3.2. Nuclear Magnetic Resonance (NMR)

As shown in Figure 3B–D, we found that the results of ^1^H NMR spectra are consistent with those of IR spectra, and the main peaks had no obvious change, which shows that ultrasound did not affect the functional groups of PHP.

#### 2.3.3. AFM and SEM Results

AFM results are shown in Figure 3E–G, with the increase of ultrasonic intensity, the surface roughness of the polysaccharide sample is significantly reduced. By analyzing the image with NanoScope Analysis 3.00 software, the roughness of PHP, PHP1, and PHP2 were determined to be 0.355, 0.191, and 0.143.

SEM results are shown in Figure 3H–J, PHP presents a multi–layer flaky structure, relatively scattered, with large gaps and no obvious holes on the surface. After the ultrasound treatment, PHP1 and PHP2 showed a clearer sheet–like structure with reduced layers, holes began to appear on the surface, and the edges were slightly curled. With the increase of ultrasonic intensity, the flaky structure of the polysaccharide was clear, the surface was smoother and had larger holes, with reduced layers, and no obvious sharp protrusions. Solubility also increased.

### 2.4. Activity of Ultrasonic PHP in RAW264.7

#### 2.4.1. Analysis of Viability Test of RAW264.7

The results of the viability experiments revealed that PHP and sonicated PHP1 and PHP2 exhibited no toxicity to macrophage RAW264.7 (Figure 4A–C). The macrophages grew best at 250 μg/mL concentrations of PHP, PHP1, and PHP2 with proliferation rates of 20.9%, 30.15%, and 33.62%, respectively. The test drug concentration was selected as 250 μg/mL for subsequent cell experiments.

#### 2.4.2. Analysis of NO Secretion Test and iNOS Expression of RAW264.7

The bactericidal effect of the macrophages is reflected in the amount of NO released in their activated state [28]. The results indicated (Figure 4D) that the content of NO released from macrophages RAW264.7 treated with PHP, PHP1, and PHP2 was significantly higher than that of the blank and negative control groups. Further, the amount of released NO increased with increasing ultrasonic intensity. The best results were obtained at the ultrasonic intensity of 500 W.

Inducible nitric oxide synthase (iNOS) produces NO by interacting with superoxide free radicals to produce peroxynitrite. After stimulation of macrophage RAW264.7, the fluorescence diagram of labeled iNOS was shown in Figure 4E, and the results were consistent with the results of NO secretion. Compared with the blank group, PHP2 showed the best activity in stimulating the expression of iNOS in macrophage RAW264.7. It was again proved that PHP had the ability to stimulate macrophage RAW264.7 to secrete NO, and this effect was also enhanced with the increase of ultrasound intensity.

#### 2.4.3. Analysis of Phagocytosis Test of RAW264.7

A Fluorescence microscope was used to observe the phagocytosis of model antigen OVA by macrophage RAW264.7 (Figure 4F) and the effects of three different samples (PHP, PHP1, and PHP2) on its phagocytosis. As shown in the figure, after adding polysaccharides, the intensity of the green fluorescence was significantly higher than in the OVA group. The intensities showed the following trend: PHP2 > PHP1 > PHP. This phenomenon indicated that polysaccharides could improve the antiphagocytic effect of macrophage RAW264.7 and the ultrasound treatment could improve this effect. Further, the antiphagocytic effect was enhanced with an increase in ultrasonic intensity. Flow cytometry measurements further verified this phenomenon (Figure 4G,H). The phagocytic ratio in the sample groups was significantly higher than that in the control group (*p* < 0.05), and there were significant differences between the sample groups, among which PHP2 showed the best effects.

#### 2.4.4. Analysis of the Polarization of RAW264.7

CD80 and CD86 are co–stimulatory factors that stimulate the polarization and activation of macrophages and are important markers of M1 polarization. They play an essential role in humoral immunity by promoting inflammatory response. RAW264.7 was treated with PHP, PHP1, and PHP2, stained with anti–CD80–PE and anti–CD86–PE, and investigated by flow cytometry (Figure 4I–L). As seen in the figure, the expression of CD80 and CD86 increased significantly under the stimulation of three samples (PHP, PHP1, and PHP2) compared with the control group. Further, the expression levels were also improved with the increase in ultrasound intensity (*p* < 0.05). It can be seen that polysaccharides promoted the typing of macrophages, increased the expression of costimulatory factors on their surface, polarized them to the M1 type, and enhanced their pro–inflammatory activity. Meanwhile, ultrasound promoted the activity of polysaccharides in macrophage polarization.

#### 2.4.5. DPPH Radical Scavenging Ability

The antioxidant activity of polysaccharide was positively correlated with its solubility [29]. An alcoholic solution of DPPH is purple in color and has a strong absorption at 517 nm. When the concentration of the DPPH radical decreased in the solution, the intensity of the 517 nm band (purple color) weakened. This decrease in intensity can be used to determine the DPPH radical scavenging ability of a sample. In the experiment, the same concentration of Vc (250 μg/mL) as the positive control and the scavenging rate of DPPH radical by PHP and its ultrasonic derivatives PHP1 and PHP2 was used, as shown in Figure 4M. As a positive control, the clearance rate of Vc was high, reaching 80.3%, while the clearance rate of PHP was below 20%. However, the DPPH free radical scavenging ability of PHP1 and PHP2 was significantly improved (*p* < 0.05) compared with PHP. Between PHP1 and PHP2, the latter had a better scavenging effect. These results indicate that polysaccharides can scavenge DPPH radicals, and ultrasonic treatment enhances the effect and improves the antioxidant activity of the polysaccharides.

#### 2.4.6. Hydroxyl Radical Scavenging Ability

Hydroxyl radical scavenging was also an important indicator of antioxidant capacity, like DPPH radical scavenging. In this experiment, Vc was used as a positive control, and the results of hydroxyl radical scavenging are shown in Figure 4N. The scavenging effect of PHP2 was less than Vc but significantly higher than that of PHP and PHP1. Although there was no significant difference between PHP and PHP1, the scavenging ability of PHP1 was better than that of PHP to some extent. When the ultrasonic intensity reached 500 W (PHP2), the scavenging ability of PHP2 was significantly higher than that of PHP1 (*p* < 0.05). The results showed that ultrasonic treatment improved the hydroxyl radical scavenging ability of polysaccharides and enhanced their antioxidant activity.

### 2.5. Activity of Ultrasonic PHP In Vivo

#### 2.5.1. Blood Cell Analysis

As shown in Figure 5A,B, PHP2 significantly increased the number of white blood cells and lymphocytes compared with the Saline group. Therefore, ultrasonic treatment has the effect of enhancing the cellular immunity of polysaccharides.

#### 2.5.2. Spleen Index

As the body’s largest secondary lymphoid organ and immune organ, the spleen plays a vital role in its functioning. The spleen has no lymphatic vessels, so all antigens and cells must enter the spleen with the blood [30]. Thus, detecting changes in the spleen is an important indicator of immune function. As shown in Figure 5C, the spleen index of PHP1 and PHP2 were significantly higher than those of the Saline group and PHP, indicating that ultrasound treatment enhanced the effect of polysaccharides on the growth of spleen tissue.

#### 2.5.3. Antioxidant Activity

The impaired balance between reactive oxygen species (ROS) production and antioxidant defense is an important cause of oxidative stress in numerous diseases. Antioxidant enzymes such as superoxide dismutase (SOD) play a key role in maintaining this balance [31]. As an oxidation product, malonic dialdehyde (MDA) is widely used to detect antioxidants. As demonstrated in Figure 5D, both PHP1 and PHP2 showed significant differences compared with the control group. Further, the level of T–AOC also increased with the increase in ultrasound intensity. As shown in Figure 5E, only PHP2 significantly differed from the blank group. There was no significant difference between the experimental groups, with PHP2 only showing an upward trend to a certain extent. As shown in Figure 5F, there were significant differences in MDA content among the groups, and the most obvious difference was in PHP2.

#### 2.5.4. Histological Evaluation

In order to determine the biosafety of PHP2, HE staining was used in the study to conduct pathological examination of mouse organs. As shown in Figure 5G, compared with the blank group, no physiological damage was found in the heart, liver, spleen, lungs, or kidneys of mice in PHP2 group, indicating that PHP2 was not toxic to organs and had good biosafety.

## 3. Materials and Methods

### 3.1. Plant Materials and Chemicals

*Pesudostellariae radix* was purchased from Sichuan Zangxitang Biotechnology Co., Ltd. (Chengdu, China). Commercial assay kits for determining protein content, superoxide dismutase (SOD), Total antioxidant capacity (T-AOC), and malondialdehyde (MDA) were obtained from the Nanjing Jiancheng Bioengineering Research Institute (Nanjing, China). D–galactose and dextrose standard samples were obtained from Chengdu kelong chemical reagent factory (Chengdu, China). Phenol, sodium chloride, trifluoroacetic acid, sulfuric acid, hydrochloric acid, sodium hydroxide, and trichloroacetic acid were obtained from Kelong Chemical Co., Ltd. (Chengdu, China). DEAE cellulose–S28556 was obtained from Yuanye Biotechnology Co., Ltd. (Shanghai, China). Trifluoroacetic acid was obtained from Sinopharm Chemical Reagent Co., Ltd. (Shanghai, China). DMEM, Cell Counting Kit–8 (CCK8), and sterile PBS were obtained from Boster Biological Technology Co. Ltd. (Wuhan, China). L–Ascorbic acid was purchased from Yi En Chemical Technology Co., Ltd. (Shanghai, China). The Nitric Oxide Assay kit was obtained from Yuhengsheng Material Technology Co., Ltd. (Suzhou, China). Neutral red was purchased from Leagene Biotechnology Co., Ltd. (Beijing, China). LPS was purchased from Solarbio Technology Co., Ltd. (Beijing, China), and Hydrocortisone sodium succinate (HCSS) was purchased from National Institutes for Food and Drug Control (Beijing, China).

### 3.2. Extraction Method of PS

#### 3.2.1. Quantification of PS via Standard Curve Method

The total PS content of the extract was assessed by phenol–H_2_SO_4_ procedure, and the standard curve was established using D–glucose as a standard [32]. First, 5mL of phenol was accurately measured, diluted with distilled water up to 100 mL in a volumetric flask, and filtered to obtain a 5% phenol solution. The prepared phenol 5% phenol solution was stored in the dark. Next, the glucose was oven–dried to constant weight at 105 °C. A total of 100 mg of glucose was accurately weighed using an electronic balance and diluted with distilled H_2_O in a 100 mL volumetric flask. The 10 mL aliquot of this solution was diluted 5 times, yielding a 0.2 mg/mL standard glucose solution. Finally, 0, 0.1, 0.2, 0.4, 0.6, and 0.8 mL of glucose standard solution was pipetted in the test tubes and mixed with 2 mL of distilled water, 1 mL of 5% phenol solution, and 5 mL c.c. H_2_SO_4_ (carefully added). The mixture was shaken and allowed to stand for 10 min. These samples’ optical density (OD) was measured at 490 nm. A standard curve was prepared from the equation, Y = 0.013X + 0.0271, R^2^ = 0.9993, where Y represents the OD_490_ and X is the concentration of the glucose solution in μg/mL. The LOD value of this standard curve was calculated as 0.22278 and the LOQ value as 0.66836. This standard curve was used to determine the sugar content in all extracts.

#### 3.2.2. Measurement of the PS Content

An accurate method for PS quantification is necessary to determine the extraction rate of PS. A total of 1 mL of the extracted PS solution was weighed into a test tube and mixed with 2 mL of distilled water. Next, 1 mL of 5% phenol solution and 5 mL c.c. H_2_SO_4_ was slowly added to the mixture. The mixture was shaken and allowed to settle for 10 min. The PS content of *P. radix* extract was determined using a previously established standard curve, and the extraction rate was calculated as follows:Extraction rate (%) = X × V/(M × 10^6^) × 100%,
where X is the concentration of the sample solution (μg/mL), V is the volume (mL), and M is the weight (g) of the *P. radix* powder.

#### 3.2.3. Single Factor Experiment and Orthogonal Experimental Design

Single–factor experiments were performed to examine the effect of several experimental variables on the extraction rate of PS from *P. radix*. *P. radix* (100.0 g) was washed with distilled water, oven–dried in a GZX–9140 MBE drying oven (Shanghai boxun medical biological instrument Co., Ltd., Shanghai, China), ground with an FD–15–T–350A grinder (Yongkang Fendou Industry Trade Co., Ltd., Jinhua, China), and filtered through a 60–mesh sieve. A total of 5 g of dry *P. Radix* powder was put in a 200 mL beaker and mixed with a known amount of distilled water. A water bath controlled the extraction temperature. After extraction, the solution was filtered with the gauze, and the solid residue was re–extracted using the previously described experimental procedure. The extracts were collected and concentrated with a rotary evaporator (Shanghai Yarong Biochemical Instrument Factory, Shanghai, China). The Sevage reagent was used for protein removal, and the process was repeated several times until a protein–free solution was obtained. Next, a 3–fold volume of absolute ethanol was added to the aqueous extract to precipitate PS. The resulting solids were separated by centrifugation (TDL–4, Wenzhou Biaonuo Instrument Co., Ltd., Wenzhou, China) at 3500× *g* for 15 min at 25 °C. The precipitates were dissolved up to 50 mL. The absorbance (i.e., OD) was recorded at 490 nm, and the PS content and its extraction rate were calculated according to the equations given in Section 3.2.2.

The extraction procedure was optimized by varying the following experimental conditions: (a) extraction times: 1–3 repeated extractions; (b) solid–liquid ratio from 1:5 to 1:25 g/mL; (c) extraction time: 1–4 h; (d) extraction temperature: 40–80 °C; and (e) ethanol concentration: 60–95%.

The extraction process of *P. radix* PS was optimized by an L_9_ (3^4^) orthogonal test using the findings of the single–factor experiments. The water bath temperature, the number of repeated extractions, and the solid–liquid ratio were chosen as factors. The level of orthogonal factors is shown in Table 2.

#### 3.2.4. Purification of PS

A total of 5 g of PS sample was dissolved in water. It was purified using a DEAE cellulose–s28556 column. The protein was eluted from the column with a peristaltic pump (DDBT–301, Zhixin Instrument Co., Ltd., Shanghai, China) at a flow rate of 1 mL/min. The eluents were added in the following sequence: pure water, 0.1 M, 0.3 M, and 0.5 M NaCl solutions. The product (10 mL/tube) was collected by a computerized automatic fraction collector [33] (BSZ–160F, Shanghai Jingke Industrial Co., Ltd., Shanghai, China). Elution curves were drawn by the phenol–sulfuric acid method. After lyophilizing the eluted product with the highest polysaccharide content, the purified polysaccharide PHP was obtained.

### 3.3. Characterization of PHP

#### 3.3.1. Infrared Spectroscopy (IR) Analysis

PHP was placed in a mortar, mixed with KBr powder, and ground uniformly. After compaction, the sample’s infrared spectrum was recorded in the 4000–400 cm^−1^ wavenumber range [34].

#### 3.3.2. Gel Permeation Chromatography (GPC) Analysis

A known amount of the sample was dissolved in ultrapure water and stirred overnight. The sample solution was filtered using a disposable microporous membrane (pore size 0.22 μm), and the filtrate was analyzed by GPC (Wyatt Technologies, Goleta, CA, USA). An OHpak column SB–806 (8 mm × 300 mm, 13 μm) connected in a series with SB–804 (8 mm × 300 mm, 10 μm) was used, and the column compartment was thermostated at 40 °C. The mobile phase was water mixed with 0.02% NaN_3_. The sample injection volume was 500 μL. The flow rate was maintained at 1 mL/min during the purification.

#### 3.3.3. Liquid Chromatography (LC) Sample Preparation

The monosaccharide content of PHP was analyzed using previously reported LC methods with slight modifications [35]. A certain amount of sample was weighed. Next, zinc acetate solution (5 mL), potassium ferrocyanide solution (5 mL), and water were slowly added to the sample in a 100 mL Erlenmeyer flask. The flask was shaken at room temperature for 1 h, centrifuged, filtered with dry filter paper, and diluted with water to the desired volume. The PHP sample (2 mL) was hydrolyzed with 1 mL trifluoroacetic acid solution (TFA, 4 M) in a sealed flask at 120 °C for 120 min. The hydrolysate was dried at 70 °C using a nitrogen blower in a bath. The dry hydrolyzed PHP was next derivatized by reacting with 0.5 mL of 0.5 M methanolic solution of PMP (1–phenyl–3–methyl–5–pyrazolone) and 0.5 mL of 0.3 M NaOH in water at 70 °C for 30 min. After 30 min, the reaction mixture was cooled to room temperature, neutralized with 0.5 mL of 0.3 M HCl, and extracted 3× with CHCl_3_ to remove the excess PMP.

#### 3.3.4. Liquid Chromatography (LC) Analysis

The aqueous layer was passed through a 0.22 μm filter and analyzed by Agilent 1200 HPLC system (Agilent Technologies Inc., Santa Clara, CA, USA) with UV detection at 250 nm. A total of 10 μL of a sample was injected into a Thermo C18 column (4.6 mm × 250 mm, 5 μm, Sepax, Newark, DE, USA) thermostated at 25 °C. A mixture of PBS (0.1 M, pH 6.7) and CH_3_CN (83:17, *v*/*v*) was used as the eluent. The flow rate was 1.0 mL/min. The monosaccharide standards were dissolved in deionized water and analyzed as mentioned above. The retention time of monosaccharides in PHP is compared with the retention time of the standard and quantitative studies are conducted based on the peak value [26].

### 3.4. Preparation and Characterization of Ultrasound PHP

#### 3.4.1. Preparation of Ultrasonic PHP

An appropriate amount of PHP was dissolved in water and placed into two beakers. The samples were given ultrasonic treatment in an ultrasonic crusher with an ultrasonic intensity of 250 W and 500 W. The samples were given several cycles of ultrasound treatment for a total of 1 h. Each cycle lasted for 2 s, and an interval of 3 s was given between two cycles. The sonicated samples were lyophilized to obtain PHP1 and PHP2.

#### 3.4.2. Comparison of IR Analysis

The infrared spectra of PHP1 and PHP2 were collected and compared with that of PHP (measured in Section 3.3.1) to determine if the ultrasound treatment affected the structural properties of the polysaccharides.

#### 3.4.3. Nuclear Magnetic Resonance (NMR)

A total of 10 mg of the samples were dissolved in 1 mL D_2_O, and their ^1^H spectra were collected and analyzed using AVⅡ–600MHz Nuclear magnetic resonance spectrometer (Bruker, Fällanden, Switzerland).

#### 3.4.4. AFM and SEM Tests

Atomic force microscopy (AFM) and Scanning electron microscopy (SEM) were used in this study to investigate the effect of ultrasound treatment on polysaccharide characterization [36]. The polysaccharide solution was deposited on the surface of fresh mica and observed using AMF (Dimension ICON, Bruker, Switzerland) after completely drying. The morphological characteristics of PHP were analyzed using a scanning electron microscope (SU8220, Hitachi, Tokyo, Japan).

### 3.5. In Vivo and In Vitro Activity of Ultrasonic–Treated PHP

#### 3.5.1. Cells and Treatment

RAW264.7 was cultured in DMEM medium containing 10% fetal bovine serum at 37 °C and 5% CO_2_. The cell concentration was adjusted to 1 × 10^5^/mL and cultured in a 96–well plate after the cells adhered for subsequent cell experiments.

#### 3.5.2. Viability Test

The RAW264.7 was co–incubated with 1000, 500, 250, 125, and 62.5 µg/mL of PHP, PHP1, and PHP2 at 37 °C. Three replicates were set for each group. After 24 h, the medium in the 96–well plate was aspirated and washed with sterile PBS twice. Next, 100 µL of culture medium and 10 µL of CCK–8 were added, and the plate was cultured. The OD was measured at 450 nm every half an hour until the OD was stable. The cell viability was then calculated using the following formula:Cell viability (%) = (experiment − blank)/(control − blank) × 100%

#### 3.5.3. Nitric Oxide (NO) Secretion Test and iNOS Expression

A blank group, LPS group, PHP, PHP1, and PHP2 groups were set up with three replicates for each group. According to the results of Section 3.5.2 on cell viability, the concentrations of PHP, PHP1, and PHP2 were set to 250 µg/mL. After the cells were cultured in a 96–well plate for 24 h, the culture medium was pipetted out, and the supernatant was transferred to a 96–well plate after centrifuging. The amount of NO released was calculated according to the NO kit instructions.

Similar to the above study, macrophage RAW264.7 was cultured into 6–well plates, and after adhesion, LPS, PHP, PHP1, and PHP2 were added to co–culture. After 12 h iNOS antibody staining was performed, and the expression of iNOS in macrophages was observed by fluorescence microscopy.

#### 3.5.4. Antigen Uptake Capacity Test

A RAW264.7 macrophage suspension with a cell density of 1 × 10^5^ cells/mL was plated on a 6–well plate with round coverslips for 12 h to allow it to adhere to the plate. A prepared OVA–FITC was mixed with PHP, PHP1, and PHP2. The mixtures were added to the culture plate and incubated for 12 h. The coverslips were taken out and fixed with 4% paraformaldehyde for 30 min, stained with the DAPI reagent for 20 min, followed by staining with DID for 20 min. The samples were washed thoroughly with PBS after each staining. Finally, the coverslips were covered with 90% glycerol and observed under a fluorescence microscope (IX73P2F, Olympus, Tokyo, Japan). The rest of the cells were scraped off and examined via flow cytometry (CyFlow Cube8, Sysmex, Norderstedt, Germany).

#### 3.5.5. Effects of the Phenotypic of RAW264.7 Macrophages

The three polysaccharide samples (PHP, PHP1, and PHP2) and LPS were co–cultured with macrophages for 12 h. The cells were scraped off and stained with anti–mouse CD80+ and CD86+ antibodies and finally analyzed by flow cytometry (CyFlowCube8, Sysmex Co., Ltd., Norderstedt, Germany).

#### 3.5.6. DPPH Radical Scavenging Test

The scavenging rate of three different samples (PHP, PHP1, and PHP2) to DPPH was determined according to the DPPH radical scavenging kit instructions. The absorbance at 517 nm was measured by spectrophotometer, and the DPPH radical scavenging rate of the samples was obtained from the following relation:DPPH radical scavenging rate (%) = (1 − (experiment − control)/blank × 100%

#### 3.5.7. Hydroxyl Radical Scavenging Test

The hydroxyl radical scavenging rate of the three samples (PHP, PHP1, and PHP2) was determined according to the instructions of the hydroxyl radical scavenging kit. A blank tube, a standard tube, a control tube, and an experimental tube were set up, and the hydroxyl radical scavenging ability of the samples was obtained from the following relationship.
Hydroxyl radical scavenging ability (U/mL) = (control − experiment)/(standard − blank) × standard concentration/sampling quantity

#### 3.5.8. Animals and Treatment

Forty female Kunming mice (weighing 30 ± 2 g, 8 weeks old) were purchased from Chengdu Dashuo Laboratory Animal Center (Chengdu, China). The mice were accommodated in polypropylene boxes in a controlled–temperature room with poplar sawdust bedding, maintaining good sanitary conditions. Pathogen–free meals and water were given to the mice. The room temperature was maintained at 22 ± 1 °C, and a regular light/dark (12 h/12 h) cycle was maintained. In this study, all animal experiments were conducted in accordance with internationally recognized principles set out in the Guidelines for the Management and Use of Laboratory Animals issued by the Chinese Government and approved by Southwest Minzu University. After the adaptive feeding for 7 days, forty female mice were randomly divided into the following four groups (*n* = 10): (1) Saline, (2) PHP, (3) PHP1, (4) PHP2. All groups except the Saline group were treated consecutively during the course of the experiment (7 days, Table 3) by gavage treatment of 0.5 mL (300 mg/kg·d). The mice were examined every day.

#### 3.5.9. Sample Collection and Preparation

On the 8th day of the experiment, 3 mice were randomly selected from each group, weighed, and peripheral blood samples were collected from their retro–orbital venous plexus to detect the number of lymphocytes. The spleen was weighed to calculate the spleen index and the liver was ground into tissue homogenate. The SOD, T–AOC activities, and the MDA concentration in the tissue were determined using commercially available kits.

#### 3.5.10. Histological Evaluation

On the 8th day of the experiment, the heart, liver, spleen, lung, and kidney of mice in the blank group and PHP2 group were collected and made into sections and pathological changes were observed by HE staining to evaluate the biosafety of PHP2.

#### 3.5.11. Data Analysis

Statistical analysis was done using the SPSS program (SPSS, Version 11.5, SPSS Inc., Chicago, IL, USA). ANOVA was used to compare results between multiple groups. All values were reported as the mean ± standard deviation (SD). A 95% confidence level (*p* < 0.05) was chosen as statistically significant and indicated with one asterisk.

## 4. Conclusions

This study investigated the extraction and characterization of PHP extracted from *Pseudostellariae radix*. Further, the effect of ultrasound treatment on the properties of PHP, along with its in vitro and in vivo biological activity, was investigated. The extraction process of *P. radix* PS was optimized using a combination of single–factor experiments and the L_9_ (3^4^) orthogonal test. The optimized experimental variables were as follows: extraction temperature—50 °C, solid–liquid ratio—1:15, and extraction time—100 min. The purified polysaccharide PHP was a complex polysaccharide composed of 10 monosaccharides, of which glucose, galactose, and arabinose were the most abundant. Meanwhile, ultrasound treatment did not significantly change the structural properties of PHP. In vitro and in vivo studies showed that ultrasonic–treated PHP showed good antioxidant activity and significantly increased the ability of macrophage RAW264.7 to release NO and phagocytic antigens. Moreover, histological evaluation showed that PHP2 had good biosafety. Further, ultrasonic–treated PHP also increased the number of immune cells and promoted the spleen’s growth and the liver’s antioxidant capacity. The results of this study demonstrate that PHP has tremendous potential as a food additive to enhance antioxidant capacity and boost immunity. The results also proved that the biological activity of PHP could be enhanced by ultrasound treatment.

## Figures and Tables

**Figure 1 molecules-29-00142-f001:**
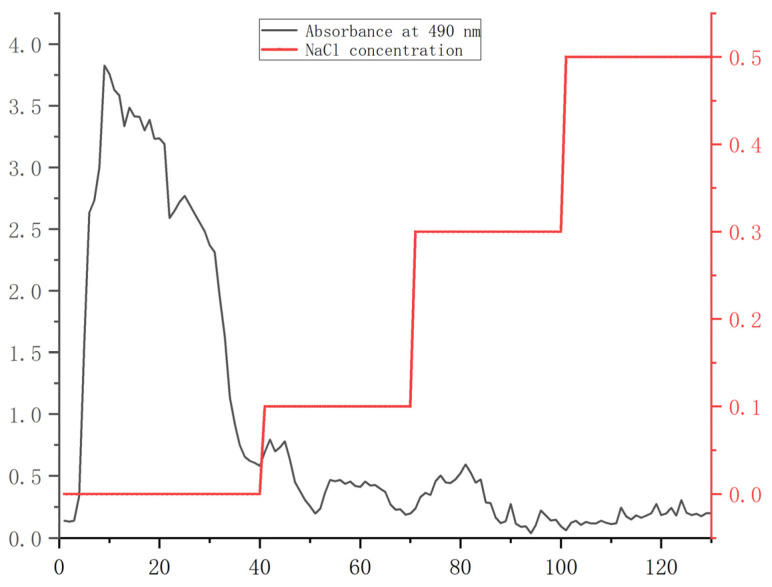
Elution curves of PS.

**Figure 2 molecules-29-00142-f002:**
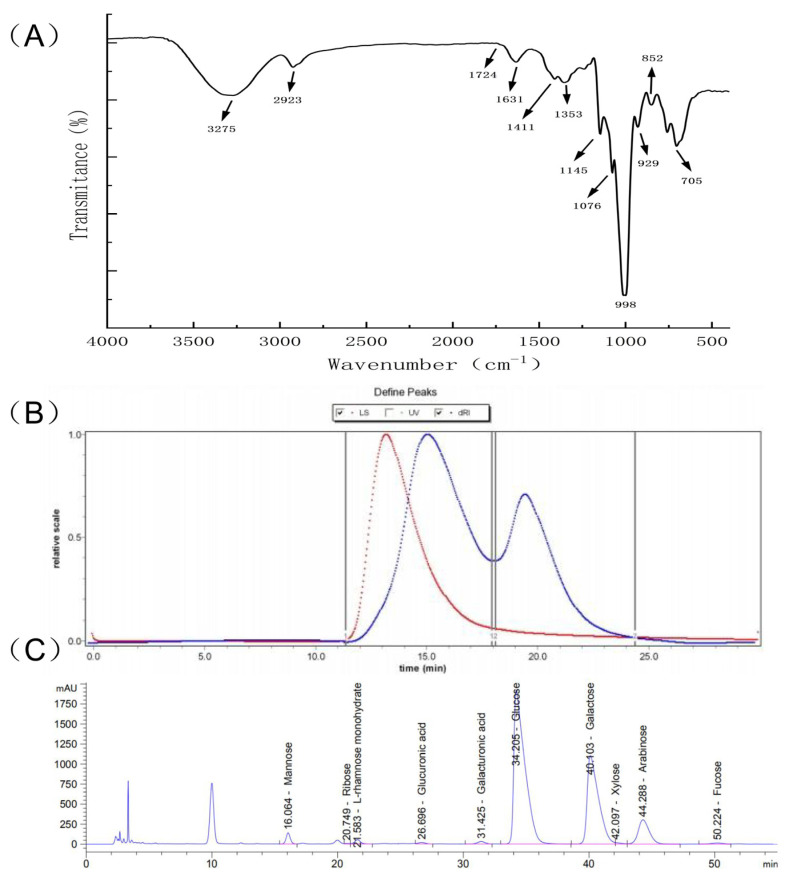
Structural representation of PHP. (**A**) The infrared spectrum of PHP, (**B**) The results of the GPC spectrum, different colors represent different molecular weight curves, (**C**) The results of liquid chromatography analysis.

**Figure 3 molecules-29-00142-f003:**
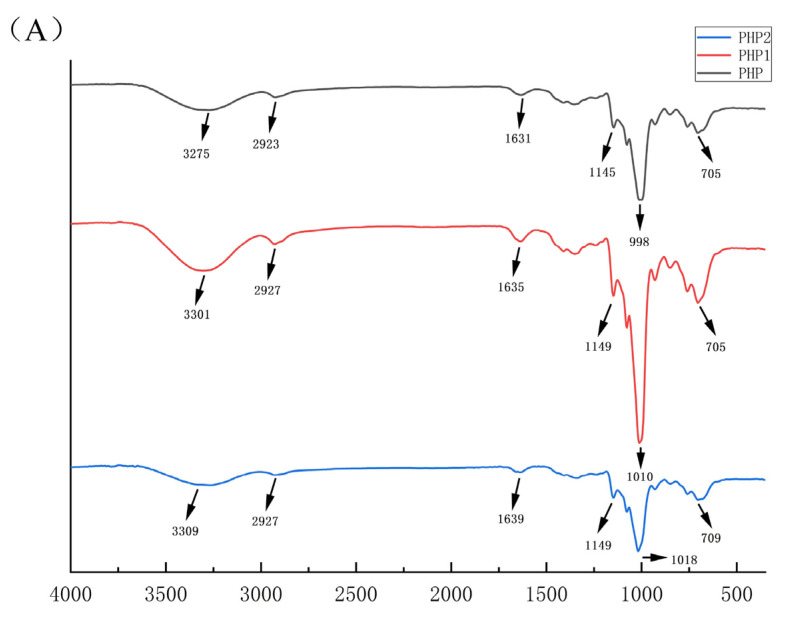
Characterization comparison chart of PHP with different ultrasonic intensities. (**A**) IR, (**B**–**D**) ^1^H NMR of PHP, PHP1, and PHP2, (**E**–**G**) Atomic force micrographs of PHP, PHP1, and PHP2, and PHP2, (**H**–**J**) Scanning electron micrographs of PHP, PHP1.

**Figure 4 molecules-29-00142-f004:**
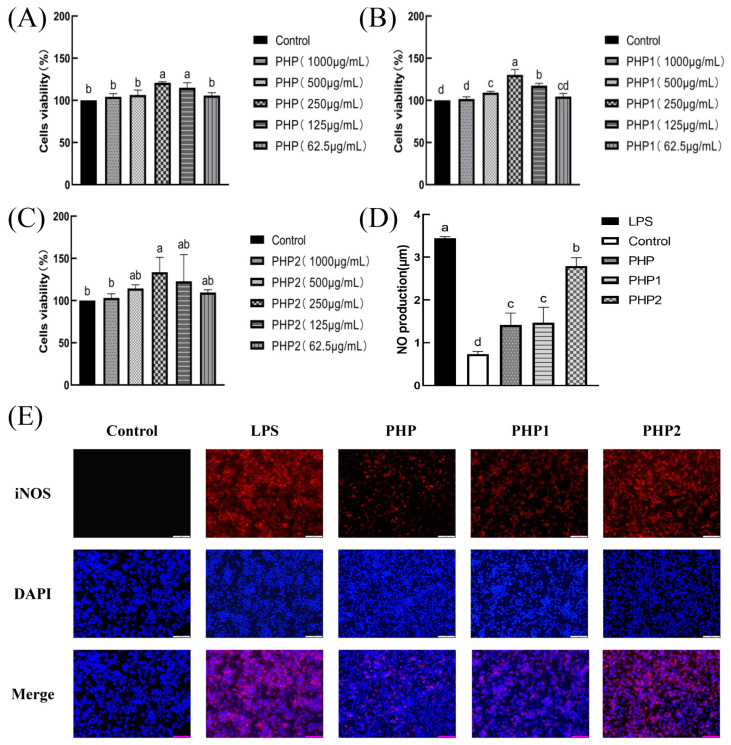
Activity of ultrasonic PHP in RAW264.7. (**A**) Effects of PHP on cell viability, (**B**) Effects of PHP1 on cell viability, (**C**) Effects of PHP2 on cell viability, (**D**) Effects of PHP, PHP1, and PHP2 on NO release from RAW264.7, (**E**) Effects of PHP, PHP1, and PHP2 on iNOS expression of RAW264.7, (**F**–**H**) Effects of PHP, PHP1, and PHP2 on cell phagocytosis. The expression frequencies of CD80+ (**I**,**K**) and CD86+ (**J**,**L**) in macrophages. (**M**) Effects of PHP, PHP1, and PHP2 on DPPH radical scavenging, (**N**) Effects of PHP, PHP1, and PHP2 on hydroxyl radical scavenging. *p* < 0.05 indicates a significant difference. Letters that are not exactly the same indicate significant differences (*p* < 0.05).

**Figure 5 molecules-29-00142-f005:**
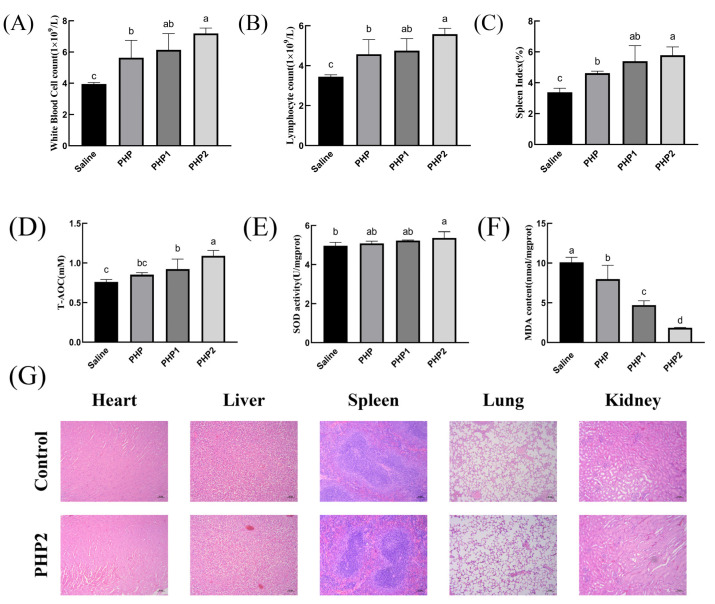
Activity of ultrasonic PHP in vivo. (**A**) White blood cell count, (**B**) Lymphocyte count, (**C**) Spleen Index, (**D**) T–AOC, (**E**) SOD, (**F**) MDA, (**G**) Histological evaluation. *p* < 0.05 indicates a significant difference. Letters that are not exactly the same indicate significant differences (*p* < 0.05).

**Table 1 molecules-29-00142-t001:** Molecular weight distribution of PHP.

Analysis Item	Index	Peak 1	Peak 2
Molecular Weight	Distribution Index PD (Mw/Mn)	2.501	1.342
	Number average molecular weight Mn (g/mol)	463,500	125,900
	Weight average molecular weight Mw (g/mol)	1,159,000	168,900

**Table 2 molecules-29-00142-t002:** Levels of three factors chosen for the optimization of extraction through the orthogonal experimental design.

Levels	Bath Temperature	Extraction Time	Material to Water Ratio
1	40	60	1:10
2	50	80	1:15
3	60	100	1:20

**Table 3 molecules-29-00142-t003:** Animal grouping and treatment.

Groups	Treatment
Saline	0.5 mL Saline
PHP	0.5 mL PHP (300 mg/kg B.W.)
PHP1	0.5 mL PHP1 (300 mg/kg B.W.)
PHP2	0.5 mL PHP2 (300 mg/kg B.W.)

## Data Availability

All data generated or analyzed during this study are included in this published article and its Appendix A.

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
