# Peer review of "Extraction of Polysaccharides from Root of Pseudostellaria heterophylla (Miq.) Pax. and the Effects of Ultrasound Treatment on Its Properties and Antioxidant and Immune Activities"

_molecules, 2023, doi:10.3390/molecules29010142_

Round 1
Reviewer 1 Report
Comments and Suggestions for Authors
I have gone through the manuscript which dealt with the hydrophilic polysaccharides (PS) extraction from the tuberous roots of important medicinal herb Pseudostellaria heterophylla using ultrasound assisted protocol followed by characterization and assessment of biological potential of PS. The work presented is quite innovative and accomplished in a systematic way. The background of the problem and problem statement along with rational of the research is well given. The experimental protocol involves the use of state-of-the-art techniques. The results and discussion is also well presented. However, I have following comments to be addressed:
Title : Th term Pseudollaria Radix used in the title does not reflect species; title can be more catchy and need to be revised as under:
Ultrasound -assisted Extraction of Polysaccharides from Root of Pseudostellaria heterophylla (Miq.) Pax.: Studying the Effects on Structural Features and Antioxidant and Immune Activities
-2.1. Plant Materials and Chemicals
How identification of plant samples was made, there is no detailed information given. For example, what was the sampling design?, did only one sample purchased then how it can be claimed as a representative sample? Also, it needs to be clarified whether this is a wild or cultivated species/population!
Results and Discussion Section: Please try to elaborate further the discussion about the effect of ultrasound -assisted extraction on structural changes of PS, and thus their link with improved biological (antioxidant and Immune) activities. There is need to focus and highlight structure-activity relationship.
Reviewer 2 Report
Comments and Suggestions for Authors
The manuscirpt deals with the extraction and characterization of PHP extracted from Pseudostellariae Radi in order to use in foods as additive. The topic is interesting and I would recommend its publication in the Molecules journal after following revisions;
- Line 51-57, ....In normal life activities, oxidation is vital for providing energy. However, when there is a problem in the metabolic pathway, the body's antioxidant system can not eliminate the excess oxygen free radicals, damaging tissues and cells (Li et al., 2020; Pattanayak et al., 2017). Recent studies have shown that bio-polysaccharides isolated from Ganoderma capense, Phyllostachys pubescens (Zhang, Wang, Yu, & Zhao, 2011), and Chuanminshen violaceus (Fan et al., 2017) have strong antioxidant activity and efficiently scavenge free radicals(Jiang et al., 2016)....
The authors should add a part in rrelation to the LC analysis of similar applications.
-Line 163-173 .....A certain amount of sample was weighed. Next, zinc acetate solution (5 mL), potassium ferrocyanide solution (5 mL), and water were slowly added to the sample in a 100 mL Erlenmeyer flask. The flask was shaken at room temperature for 1h, centrifuged, filtered with dry filter paper, and diluted with water to the desired volume. PHP sample (2 mL) was hydrolyzed with 1 mL trifluoroacetic acid solution (TFA, 4 M) in a sealed flask at 120 °C for 120 min. The 168 hydrolysate was dried at 70°C using a nitrogen blower in a bath. The dry hydrolyzed PHP was next derivatized by reacting with 0.5 mL of 0.5 M methanolic solution of PMP (1- phenyl-3-methyl-5-pyrazolone) and 0.5 mL of 0.3 M NaOH in water at 70°C for 30 min. After 30 mins, the reaction mixture was cooled to room temperature, neutralized with 0.5 mL of 0.3 M HCl, and extracted 3× with CHCl3 to remove the excess PMP...
This part should be given as a separate part such as sample preparation.
-Line 92-95 .... The 10 mL aliquot of this solution was diluted 5 times, yielding a 0.2 mg/mL standard glucose solution. Finally, 0, 0.1, 0.2, 0.4, 0.6, and 0.8 mL of glucose standard solution was pipetted in the test tubes and mixed with 2 mL of distilled water, 1 mL of 5% phenol solution, and 5 mL c.c. H2SO4....
Whole method validation parameters including LOD, LOQ should be givenç
- Line 298-301 ....The FTIR spectrum of PHP was recorded in the 4000 - 400 cm−1 range. According to Fig. 2A, the FTIR spectrum PHP matched the characteristics of the absorption spectra of polysaccharides. There was an intensive, broad absorption peak at 3275 cm...
Fig 2A is blurred and it should be enlarged
- Line 322-327 ......The monosaccharide content of PHP was determined by LC after acid hydrolysis and pre-column derivatization (Fig. 2C). The monosaccharides were identified by the retention Molecules 2023, 28, x FOR PEER REVIEW 9 of 19 times of corresponding monosaccharide standards. Galactose, glucose, arabinose, mannose, L-rhamnose monohydrate, galacturonic acid, xylose, glucuronic acid, fucose, and ribose with the molar ratios of 743.54, 718.78, 118.1, 20.91, 12.05, 9.06, 4.78, 4.77, 3.93 and 1.19 were the main constituents of PHP....
In this part, the authors should discuss their results according to chrmatographic conditions. In addtion, Fig 2C should be given as a separate figure including chromatographic conditions.
Comments on the Quality of English Language
The manuscript English needs minor revision.
Round 2
Reviewer 2 Report
Comments and Suggestions for Authors
accept
Comments on the Quality of English Languageneeds minor revisions